# Radiomics-Based Machine Learning Model for Diagnosis of Acute Pancreatitis Using Computed Tomography

**DOI:** 10.3390/diagnostics14070718

**Published:** 2024-03-28

**Authors:** Stefanie Bette, Luca Canalini, Laura-Marie Feitelson, Piotr Woźnicki, Franka Risch, Adrian Huber, Josua A. Decker, Kartikay Tehlan, Judith Becker, Claudia Wollny, Christian Scheurig-Münkler, Thomas Wendler, Florian Schwarz, Thomas Kroencke

**Affiliations:** 1Clinic for Diagnostic and Interventional Radiology and Neuroradiology, University Hospital Augsburg, 86156 Augsburg, Germany; stefanie.bette@uk-augsburg.de (S.B.); luca.canalini@uk-augsburg.de (L.C.); laura-marie.feitelson@uk-augsburg.de (L.-M.F.); adrian.huber@uk-augsburg.de (A.H.); josua.decker@uk-augsburg.de (J.A.D.); kartikay.tehlan@uk-augsburg.de (K.T.); judith.becker@uk-augsburg.de (J.B.); claudia.wollny@uk-augsburg.de (C.W.); christian.scheurig@uk-augsburg.de (C.S.-M.); thomas.wendler@uk-augsburg.de (T.W.); 2Department of Diagnostic and Interventional Radiology, University Hospital Würzburg, University of Würzburg, 97080 Würzburg, Germany; piotr.a.woznicki@gmail.com; 3Institute of Digital Health, University Hospital Augsburg, Faculty of Medicine, University of Augsburg, 86356 Neusaess, Germany; 4Computer-Aided Medical Procedures and Augmented Reality, School of Computation, Information and Technology, Technical University of Munich, 85748 Garching bei Muenchen, Germany; 5Centre for Diagnostic Imaging and Interventional Therapy, Donau-Isar-Klinikum, 94469 Deggendorf, Germany; florian.schwarz@donau-isar-klinikum.de; 6Centre for Advanced Analytics and Predictive Sciences (CAAPS), University of Augsburg, 86159 Augsburg, Germany

**Keywords:** radiomics, classification, segmentation, artificial intelligence, CT, acute pancreatitis

## Abstract

In the early diagnostic workup of acute pancreatitis (AP), the role of contrast-enhanced CT is to establish the diagnosis in uncertain cases, assess severity, and detect potential complications like necrosis, fluid collections, bleeding or portal vein thrombosis. The value of texture analysis/radiomics of medical images has rapidly increased during the past decade, and the main focus has been on oncological imaging and tumor classification. Previous studies assessed the value of radiomics for differentiating between malignancies and inflammatory diseases of the pancreas as well as for prediction of AP severity. The aim of our study was to evaluate an automatic machine learning model for AP detection using radiomics analysis. Patients with abdominal pain and contrast-enhanced CT of the abdomen in an emergency setting were retrospectively included in this single-center study. The pancreas was automatically segmented using TotalSegmentator and radiomics features were extracted using PyRadiomics. We performed unsupervised hierarchical clustering and applied the random-forest based Boruta model to select the most important radiomics features. Important features and lipase levels were included in a logistic regression model with AP as the dependent variable. The model was established in a training cohort using fivefold cross-validation and applied to the test cohort (80/20 split). From a total of 1012 patients, 137 patients with AP and 138 patients without AP were included in the final study cohort. Feature selection confirmed 28 important features (mainly shape and first-order features) for the differentiation between AP and controls. The logistic regression model showed excellent diagnostic accuracy of radiomics features for the detection of AP, with an area under the curve (AUC) of 0.932. Using lipase levels only, an AUC of 0.946 was observed. Using both radiomics features and lipase levels, we showed an excellent AUC of 0.933 for the detection of AP. Automated segmentation of the pancreas and consecutive radiomics analysis almost achieved the high diagnostic accuracy of lipase levels, a well-established predictor of AP, and might be considered an additional diagnostic tool in unclear cases. This study provides scientific evidence that automated image analysis of the pancreas achieves comparable diagnostic accuracy to lipase levels and might therefore be used in the future in the rapidly growing era of AI-based image analysis.

## 1. Introduction

Acute pancreatitis (AP) is a life-threatening disease with an increasing annual incidence [1]. Despite advances in diagnosis and therapy, severe AP still has a high morbidity and mortality [2]. Besides clinical features and blood tests, imaging (e.g., ultrasound or contrast-enhanced computed tomography (CT)) plays an essential role in the diagnostic workup of AP. According to an international consensus and the revised Atlanta classification, the diagnosis of AP requires two of the following three features: (i) abdominal pain, (ii) elevated lipase activity, (iii) consistent findings in CT [3]. CT is an important component in the primary diagnosis of AP as well as in follow-up imaging to detect complications of AP, e.g., development of necrosis and peripancreatic (infected) fluid collections, portal vein thrombosis, pseudoaneurysms or bleeding.

Texture analysis (also called radiomics) is a quantitative method of image analysis which describes the conversion of images into data and has become important for medical image analysis in the last decade. Using this method, attenuation can be quantitatively assessed in each voxel. Radiomics extracts quantitative information from each voxel and assesses the distribution of intensities (histograms) and the shape of the mask/region of interest as well as relationships between different voxels. This method allows for an objective and observer-independent image analysis to facilitate the correct diagnosis [4,5]. The number of studies analyzing the texture of organs and/or tumors rapidly increased during the past decade; these analyses showed promising results of radiomics, especially in the field of cancer imaging, but also in various other fields.

Previously, recent studies performed texture analyses of the pancreas and showed that the extraction of radiomics features from the pancreas might facilitate the diagnosis of diabetes mellitus [4], the prognostic value of pancreatic cancer [6,7] and differentiation between pancreatic lesions [8,9]. As most radiomics studies focus on malignancies, only a few studies assessed the role of texture features in inflammatory processes, e.g., acute pancreatitis [10,11]. In addition, previous studies mainly focused on the diagnosis of pancreatic lesions and the differentiation of tumors and inflammatory pancreatic diseases [11,12,13,14,15,16,17,18]. Other radiomics studies (also including MRI studies) analyzed the prognostic value of texture features for recurrence of acute pancreatitis or its severity [19,20,21,22]. A recent study assessed if radiomics features can discriminate between functional abdominal pain and acute and chronic pancreatitis [23]. The study used manual segmentation of the pancreas followed by radiomics feature extraction and showed that texture analysis is a potential tool for differentiation. It is not yet known if an automated approach using segmentation of the pancreas and radiomics-based analysis of contrast-enhanced CT of the abdomen can be used to identify patients with the presence of acute pancreatitis.

The aim of this study was to assess the value of radiomics features after automatic segmentation of the pancreas for differentiation between acute pancreatitis and non-pancreatitis in patients with abdominal pain in an emergency setting.

## 2. Materials and Methods

This retrospective single-center study was approved by the local Medical Research and Ethics Committee (MREC) (Protocol Number: 20-1153). Written informed consent was waived by the MREC due to the retrospective study design.

### 2.1. Study Population

Inclusion criteria comprised (a) the availability of contrast-enhanced abdominal CT (portal venous phase) on a second-generation dual-source MDCT scanner (SOMATOM Definition Flash, Siemens Healthineers, Erlangen, Germany ), (b) sufficient image quality (i.e., patients with blurred images or strong artifacts due to metal implants in the spine were excluded), (c) diagnosis of acute pancreatitis according to the revised Atlanta classification [3] and (d) age ≥18 years.

For the control group, the local database was searched for contrast-enhanced CT imaging of the abdomen in the portal venous phase performed using the same scanner due to unclear (upper) abdominal pain without the ICD-code K85 (acute pancreatitis) between January 2016 and November 2020.

For all patients (pancreatitis and control group), lipase activity levels were collected and shown in U/L.

### 2.2. Scanning Protocol

All patients underwent a contrast-enhanced CT of the (upper) abdomen with a second-generation dual-source MDCT scanner (SOMATOM Definition Flash, Siemens Healthineers, Erlangen, Germany) as routine clinical acquisition using a monophasic contrast injection protocol in the portal venous phase. A contrast bolus of 120 mL (Imeron 350 mgI/mL, Bracco Imaging Deutschland GmbH, Konstanz, Germany) was injected via an antecubital vein (flow rate 4.0 mL/s) and followed by a saline bolus of 30 mL. Images were acquired after a fixed delay of 75 s after contrast injection. Each patient was scanned craniocaudally in a supine position. We further used the following technical parameters: Care kV 7, 120 kV tube voltage, 120 mAs, 0.5 s rotation time, 128 × 0.6 mm collimation, pitch factor 1.0.

### 2.3. Automatic Segmentation, Feature Extraction and Selection and Statistical Analysis

Automatic segmentation of the pancreas was performed using the open-source software TotalSegmentator (version 1.5.6 [24]) in Python (version 3.7). Figure 1 shows an example of automatic segmentation of the pancreas.

For the visualization of automatic segmentation, the software 3D Slicer (version 5.2.2) was used (http://www.slicer.org (accessed on 19 July 2023), [25]). Automatic segmentation was verified by a board-certified radiologist in randomly selected cases. Radiomics features were extracted using the software package Pyradiomics (version 3.1.0, [26]). In total, 104 features were extracted (first-order, shape, Gray Level Co-occurrence Matrix [glcm], Gray Level Dependence Matrix [gldm], Gray Level Run Length Matrix [glrlm], Gray Level Size Zone Matrix [glszm], Neighbouring Gray Tone Difference Matrix [ngtdm]). First, feature normalization was performed in Python (version 3.10) using the Z-score method. Second, all extracted and normalized features were loaded into a statistical software (R Statistics, version 4.3.1, R Core Team, Vienna, Austria) [27]. Then, data were divided into a training and a test cohort using an 80/20 split. Data were analyzed and visualized using RStudio (version 2023.06.2 [28]). Unsupervised hierarchical clustering of normalized radiomics features was performed using the package ComplexHeatmap in R and Rstudio. To select the most important radiomics features, the established Boruta package was applied in R in the training cohort using a random forest (RF) feature selection. We chose the Boruta package as this is a well-established and powerful method for feature selection and its applicability has already been proven in previous radiomics studies [29]. After feature selection, all features that were confirmed as important (and/or lipase levels) were included in binary logistic regression analysis in R using the presence of pancreatitis as the dependent variable and radiomics features (and/or lipase levels) as independent variables. The model was trained on the training cohort using 5-fold cross-validation and tested on the test cohort. Receiver operating characteristic (ROC) curves and areas under the ROC curve (AUC) including 95% confidence intervals were calculated in R.

Features that were confirmed as the most important features after RF feature selection were visualized in boxplots. Mann–Whitney U tests were performed to compare all confirmed features between the two groups (acute pancreatitis vs. control). Post hoc Bonferroni correction was applied to correct for multiple testing. A *p*-value < 0.05 was considered to indicate statistically significant differences.

## 3. Results

### 3.1. Patient Cohort

A total of 267 patients with the ICD-code K85 “acute pancreatitis” between January 2016 and December 2020 and CT of the (upper) abdomen were included in the first analysis. Patients were excluded due to the following reasons: missing contrast agent (n = 24), missing portal venous contrast phase (n = 36), missing diagnosis of acute pancreatitis according to the criteria of the revised Atlanta classification [3] (n = 65), status after pancreatic surgery (n = 2), image acquisition using a different CT scanner (n = 3). Therefore, 137 patients (43 female assigned at birth, mean age at diagnosis 59.4 years [±16.7]) with a diagnosis of acute pancreatitis were included in the final study cohort.

A total of 745 patients with CT of the abdomen and the excluded diagnosis of “acute pancreatitis” were analyzed. Patients were excluded due to the following reasons: analysis using a different CT scanner (n = 374) or other symptoms/prior history (e.g., trauma, tumor or gastrointestinal bleeding, n = 233). Therefore, the final control group consisted of 138 patients (46 female assigned at birth, mean age 61.1 years [±17.9]) with contrast-enhanced CT of the abdomen due to acute abdominal pain.

Acute pancreatitis was classified as interstitial edematous AP in 111 of 137 patients [81.0%]) and as necrotizing AP in 26 of 137 patients [19.0%]). The etiologies of AP were alcohol abuse (47/137 patients [34.3%]) and biliary diseases (37/137 patients [27.0%]). Further causes comprised prior endoscopic retrograde cholangiopancreatography (ERCP) (16/137 [11.7%]) and other rare causes (e.g., autoimmune pancreatitis). Lipase levels were available for 127/137 patients with AP and for 120/137 patients in the control group. Lipase levels were elevated in the AP group (median 385 U/l [interquartile range (IQR) 124-600]) compared to the control group (24.5 U/L [15,16,17,18,19,20,21,22,23,24,25,26,27,28,29,30,31,32]). A total of 25 of 137 patients in the AP group (18.2%) presented with prior stenting of the common bile duct. The main clinical symptoms in the control group were acute abdomen and abdominal pain (94/138, [68.1%]). There was a similar distribution of patients receiving oral contrast agent prior to contrast-enhanced CT of the abdomen in the AP group (51/137, [37.2%]) and the control group (54/138, [39.1%]).

Baseline patient characteristics are shown in Table 1.

### 3.2. Cluster Analysis

Unsupervised hierarchical clustering was performed for all extracted and normalized radiomics features in patients with and without acute pancreatitis. Data are visualized in a heatmap (Figure 2).

### 3.3. Radiomics Feature Selection

Twenty-eight features were confirmed to be important features, as shown in Figure 3. Detailed data about feature importance are presented in Supplemental Appendix A. Especially, shape features (7/28) and first-order features (7/28) were selected as important features; shape features describe the shape of a given mask (segmentation)/region of interest and therefore the boundaries of the pancreas. First-order features describe the distribution of intensities (Hounsfield units in this study) for each voxel. Also, glszm features (5/28) and gldm features (5/28) were confirmed to be important features for discrimination between AP and controls. Both feature types analyze gray levels; glszm features focus on gray-level zones and gldm features on gray-level dependencies. As the three most important features for discrimination between AP and controls, “shape_SurfaceVolumeRatio”, “gldm_DependenceNonUniformity” and “shape_MeshVolume” were selected (Figure 4). Using non-parametric tests, significant differences between patients with and without AP were found in these three features (*p*-value < 0.001) and in most other selected radiomics features. Median values of all features as well as *p*-values are presented in Table 2.

### 3.4. Logistic Regression Model

All features that were confirmed as important features after RF-based selection were included in the model as independent variables and analyzed in the binary logistic regression model for AP as the dependent variable. ROC analysis of the test set revealed an excellent performance of the model, including radiomics features only for the diagnosis of AP with an average AUC of 0.932 (95%-CI: 0.852–1.00) (Figure 5). Including lipase levels only, we observed an AUC of 0.946 (95%-CI: 0.883–1.00) for the correct diagnosis. Including a combination of both—lipase levels and radiomics features—showed an excellent AUC of 0.933 (95%-CI: 0.850–1.00) for the detection of AP.

## 4. Discussion

This study showed in a large patient cohort that automatic segmentation of the pancreas followed by radiomics extraction can predict the presence of acute pancreatitis with high accuracy. This algorithm almost achieves the high diagnostic accuracy of lipase levels. Therefore, the application of this algorithm might support radiologists and clinicians to confirm the diagnosis, especially in unclear cases. Further, this study provides scientific evidence that automatic image and texture analysis of the pancreas achieves comparable diagnostic accuracy to lipase levels in the diagnostic workup of AP.

Acute pancreatitis has been increasing over the past decades. Most patients present with abdominal pain which radiates around the back, like a belt. If patients have typical symptoms and elevated lipase levels in blood tests, further CT imaging is not recommended [3,30]. In unclear cases or if complications are suspected, either ultrasound or contrast-enhanced CT of the upper abdomen is performed, depending on the assessability of ultrasound and disease severity [30]. In our study, both the acute pancreatitis group and the control group included patients with unclear abdominal pain and/or suspicion of complications, which was the basis of the indication for contrast-enhanced CT of the abdomen.

Automatic segmentation of the pancreas was performed using TotalSegmentator [24], a freely available and robust model for the segmentation of anatomic structures in CT images. This recently released software enables accurate segmentation based on a nnU-Net-segmentation algorithm and was trained on over 4000 CT scans, outperforming other segmentation models with a high Dice score [24]. The extraction of radiomics was performed using the established package “Pyradiomics” [26]. To reduce the number of features, the previously presented and well-known Boruta package was applied for feature selection [29,31]. The logistic regression model showed a very high accuracy with an AUC of 0.932 for the diagnosis of acute pancreatitis in patients with unclear abdominal pain. These results showed that quantitative image analysis can identify changes in the texture of the pancreatic parenchyma very precisely and might help radiologists and clinicians in the diagnosis of unclear cases. Similar results were also presented in a previous study [23], showing an AUC of 0.91 for the correct diagnosis. The latter study included in total 56 patients and analyzed patients with chronic pancreatitis, who were excluded from our cohort. Mashayekhi et al. mainly identified glcm features as the most important radiomics features which are generated by the intensity of voxel pairs [23,32]. In contrast, our study identified features from different sections, mainly shape and first-order features that describe the distribution of voxel intensities [32]. The differences in these features are quite conceivable, as the pancreas changes in shape and intensity during acute pancreatitis, showing more blurred boundaries during inflammation as well as reduced contrast enhancement in cases of necrosis. Both organ changes are also used as diagnostic criteria for AP in contrast-enhanced CT by radiologists; however, in some cases, these changes might be very subtle and not visible to the human eye. These cases might be of high relevance for quantitative image analysis and radiomics features as they may also detect these slight changes. Further studies assessing this issue are necessary to confirm the value of radiomics in cases with only subtle changes in contrast-enhanced CT images. Also, other radiomics features (e.g., glszm- and gldm features, considering gray level zones and dependencies) were shown to be important features for differentiating between acute pancreatitis cases and controls. These features are not clearly visible to the human eye and are reserved for computer-based analyses. This suggests that texture analysis provides a more profound look at these images, allowing access to data that are invisible to humans.

Previous studies that analyzed the texture of the pancreas mainly focused on the differentiation between mass-forming pancreatitis and pancreatic tumors, as this is often a major radiological challenge [12,13,14,15,16,17,18,33]. The diagnosis of AP is mainly made according to clinical symptoms and the presence of elevated lipase levels in blood tests [30]. Therefore, the value of automatic AP detection in CT images might be questionable. However, the present study included only patients with an indication for contrast-enhanced CT (either in unclear cases or with suspected complications of AP) and patients with unclear abdominal pain in the control group, representing a typical diagnostic challenge in emergency radiology. Especially in patients with unclear abdominal pain and in need of CT, the automatic detection of AP might help radiologists and clinicians in the diagnostic workup and might also accelerate the diagnosis if implemented automatically. And, regarding the current rapid increase of artificial intelligence-based solutions for automated disease detection, texture analysis might be important in the near future. This study provides proof that texture analysis has a comparable diagnostic accuracy to well-established blood tests. In addition, this study also evaluated the benefits of using both lipase levels and quantitative imaging features for AP detection. Whereas lipase levels, a well-established predictor, had a very good AUC of 0.946 (95%-CI: 0.883–1.00) for the correct diagnosis, the usage of the radiomics model also achieved a high diagnostic accuracy (0.932 (95%-CI: 0.852–1.00).

The fact that texture analysis of the pancreas is an important predictor of AP prognosis confirms its significance in the initial diagnosis of patients with CT indication. Previous studies pointed out the relevance of radiomics analyses in the prediction of AP recurrence [19,34,35]. Furthermore, the prediction of AP severity is of high clinical relevance. AP has a high morbidity and mortality and a high risk of organ failure. Early determination of disease severity would be of high clinical relevance, suggesting closer monitoring of patients with elevated risk. Recent studies showed MRI texture analysis is a potential predictor of AP severity [20,21,22]. Very recent studies also highlighted the value of CT-based texture analysis for the prediction of the prognosis and severity of AP [36,37]. Liu et al. established a clinical radiomics model, evaluated its performance in the prediction of organ failure, intensive care unit stays and the need for interventions and divided patients into low- and high-risk groups. They found that patients in the high-risk group had significantly higher rates of organ failure and longer duration of hospitalization [37]. A combination of both radiomics features and clinical features/blood tests might therefore improve patient care, including closer monitoring of high-risk patients and therefore potentially earlier detection of complications and reduction in hospitalization.

This study has limitations. First, a retrospective study design is a major limitation. Patients with unclear abdominal pain and/or suspected complications of either pancreatitis or other abdominal diseases were included. Our aim was to analyze a realistic clinical scenario which also involves patients with unclear abdominal symptoms. Second, this was a single-center study that evaluated a standardized CT protocol. Radiomics analyses strongly depend on CT protocols, contrast phases and image postprocessing; therefore, a generalization of the results for other CT scanners and protocols cannot be performed. Radiomics analyses are prone to overfitting, as a large number of data are assessed. To avoid overfitting, feature selection is performed to reduce the number of data to a minimum. This study used the well-established Boruta package for feature selection, which had been employed in previous radiomics studies. There are also other approaches for feature selection, e.g., Lasso regression. Further multi-center studies are necessary to evaluate the transferability of the results. Third, it is important to mention that this AI model might be of limited value in clinical routine. The majority of cases with AP have typical clinical symptoms and elevated lipase levels and therefore, no CT is recommended for primary diagnosis. Contrast-enhanced CT is performed either in unclear cases or for suspected complications of AP. Therefore, this model might be limited to a small group of unclear cases, either in patients with typical clinical symptoms and normal lipase levels or vice versa. Further, there is also a rare number of cases with only subtle changes in CT imaging that might benefit from this quantitative model. However, with the increasing use of AI-based detection of pathological changes, this study provides evidence that texture-based analysis provides comparable diagnostic confidence to blood tests and may therefore be implemented in automated detection in the future.

## 5. Conclusions

Automatic segmentation of the pancreas and quantitative image analysis have a high diagnostic accuracy for AP in contrast-enhanced CT and almost achieve the values of lipase levels, a well-established AP predictor. Therefore, in an emergency setting and especially in unclear cases, radiomics features might help clinicians and radiologists in the diagnostic workup. This study provides scientific evidence that automated image analysis of AP achieves comparable diagnostic confidence to blood tests and might therefore be used in the future in the rapidly growing era of AI-based image analysis.

## Figures and Tables

**Figure 1 diagnostics-14-00718-f001:**
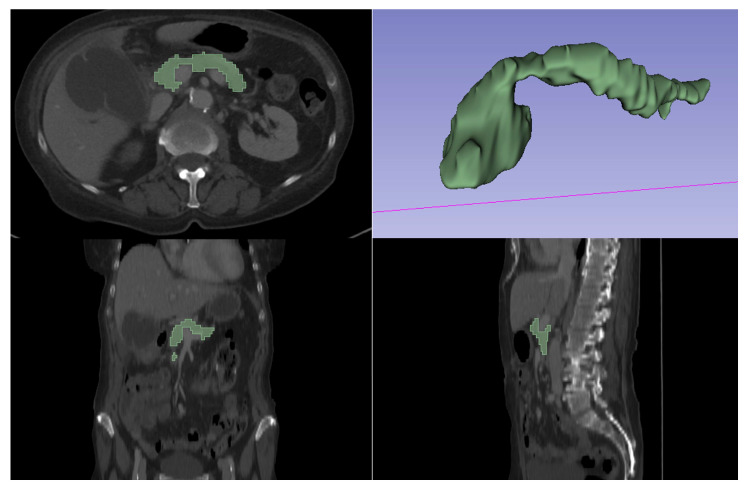
Example of automatic segmentation of the pancreas.

**Figure 2 diagnostics-14-00718-f002:**
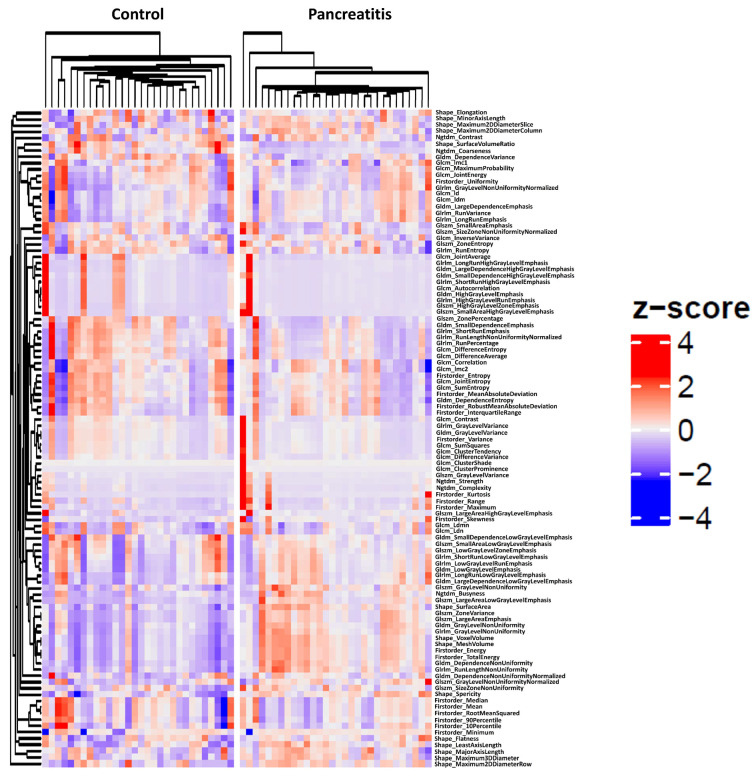
Heatmap for unsupervised hierarchical clustering of standardized radiomics features in 30 randomly selected patients with acute pancreatitis and controls.

**Figure 3 diagnostics-14-00718-f003:**
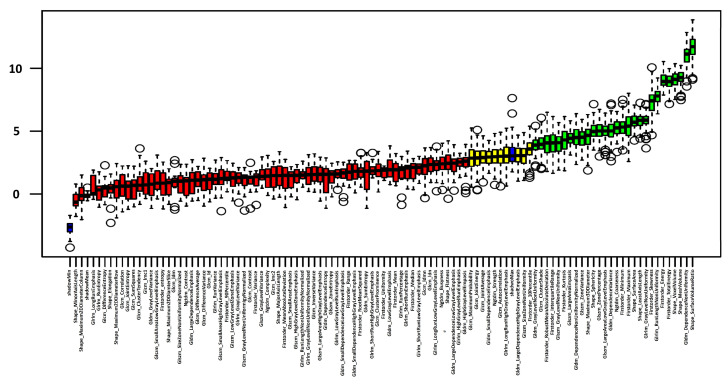
Feature selection using the random forest-based Boruta package. Confirmed features are shown in green, tentative features in yellow, rejected features in red and “shadow” features in blue. Circles show outliers and the whiskers indicate minimum and maximum.

**Figure 4 diagnostics-14-00718-f004:**
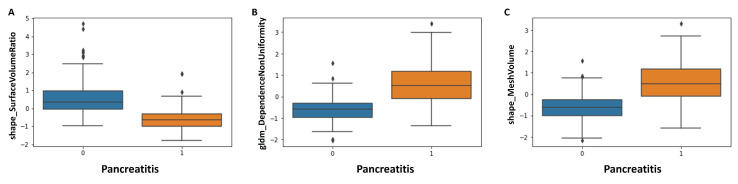
Boxplots for the three most important features for comparison between patients with and without acute pancreatitis (AP, blue without AP, orange with AP; (**A**): shape_SurfaceVolumeRatio, (**B**): gldm_DependenceNonUniformity, (**C**): shape_MeshVolume). Data shown after feature normalization (z-score) for shape_surfaceVolumeRatio, gldm_DependenceNonUniformity and shape_MeshVolume.

**Figure 5 diagnostics-14-00718-f005:**
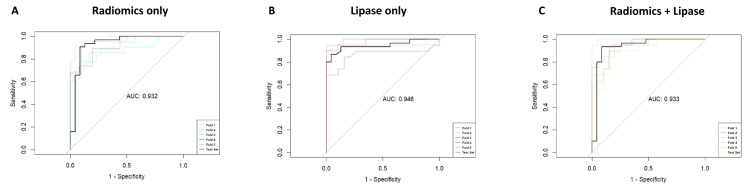
Receiver operating characteristic (ROC) curves of the logistic regression models. Areas under the ROC curve (AUC) shown for performance of the models on a test set (n = 55) for radiomics only (**A**), for lipase only (**B**) and for a combined approach using radiomics and lipase (**C**).

**Table 1 diagnostics-14-00718-t001:** Baseline patient characteristics.

	AP (n = 137)	Control (n = 138)
Sex, female (%)	43/137 (31.4%)	46/138 (33.3%)
Age, mean (± sd)	59.4 (±16.7)	61.1 (±17.9)
**AP**		
− Interstitial edematous	111/137 (81.0%)	n.a.
− Necrotizing	26/137 (19.0%)	n.a.
**Etiology of AP**		
Alcohol	47/137 (34.3%)	n.a.
Biliary	37/137 (27.0%)	n.a.
Post-ERCP	16/137 (11.7%)	n.a.
Others	22/137 (16.1%)	n.a.
**Clinical symptoms control group**		
Acute abdomen	n.a.	32/138 (23.2%)
Upper abd. pain	n.a.	35/138 (23.4%)
Lower abd. pain	n.a.	27/138 (19.6%)
Abd. infection	n.a.	11/138 (8.0%)
Others	n.a.	33/138 (23.9%)
Oral contrast agent	51/137 (37.2%)	54/138 (39.1%)
Common bile duct stenting	25/137 (18.2%)	n.a.
Lipase (U/L), median (IQR)	385 (124–600)	24.5 (15–32)

Normally distributed data shown as mean (±sd), non-normally distributed data shown as median (interquartile range, IQR). AP: acute pancreatitis; ERCP: endoscopic retrograde cholangiopancreatography. n.a.: not applicable.

**Table 2 diagnostics-14-00718-t002:** Quantitative analysis of selected and normalized radiomics features.

Feature	Pancreatitis	Control	*p*-Value
shape_LeastAxisLength	0.51	−0.51	<0.001
shape_Maximum3DDiameter	0.35	−0.24	0.084
shape_MeshVolume	0.47	−0.63	<0.001
shape_Sphericity	0.34	−0.41	<0.001
shape_SurfaceArea	0.50	−0.50	<0.001
shape_SurfaceVolumeRatio	−0.63	0.34	<0.001
shape_VoxelVolume	0.47	−0.63	<0.001
firstorder_Energy	0.48	−0.56	<0.001
firstorder_InterquartileRange	−0.01	−0.10	1.000
firstorder_Kurtosis	−0.25	−0.22	0.245
firstorder_Maximum	−0.33	−0.30	1.000
firstorder_Minimum	0.35	0.20	0.013
firstorder_Skewness	−0.06	−0.21	0.207
firstorder_TotalEnergy	0.48	−0.56	<0.001
glcm_ClusterShade	0.03	0.02	0.179
glrlm_GrayLevelNonUniformity	0.39	−0.57	<0.001
glrlm_RunLengthNonUniformity	0.43	−0.54	<0.001
glszm_GrayLevelNonUniformity	0.36	−0.51	<0.001
glszm_LargeAreaEmphasis	0.33	−0.63	<0.001
glszm_LargeAreaLowGrayLevelEmphasis	0.14	−0.58	<0.001
glszm_ZonePercentage	−0.58	0.22	<0.001
glszm_ZoneVariance	0.33	−0.63	<0.001
gldm_DependenceNonUniformity	0.52	−0.60	<0.001
gldm_DependenceNonUniformityNormalized	−0.06	−0.37	0.221
gldm_DependenceVariance	−0.15	0.31	0.008
gldm_GrayLevelNonUniformity	0.38	−0.53	<0.001
gldm_LargeDependenceHighGrayLevelEmphasis	−0.34	−0.26	0.113
ngtdm_Coarseness	−0.46	0.05	<0.001

Data shown as median; *p*-value shown after Bonferroni correction; firstorder: describe distribution of voxel intensities; shape: describe the shape of the mask; glcm: describe the probability function; glszm: quantify gray level zones; gldm: quantify gray level dependencies; glrlm: quantify length in number of pixels; ngtdm: quantify gray level differences between neighboring voxels.

## Data Availability

The data presented in this study are available on request from the corresponding author.

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
