# Peer review of "Radiomics-Based Machine Learning Model for Diagnosis of Acute Pancreatitis Using Computed Tomography"

_diagnostics, 2024, doi:10.3390/diagnostics14070718_

Round 1

Reviewer 1 Report

Comments and Suggestions for Authors

Congratulations!

The presented study is  evaluating an automatic machine learning model for diagnosis of acute pancreatitis using radiomics analysis. The enroledpPatients with abdominal pain and contrast-enhanced CT of the abdomen in an emergency setting. The logistic regression model showed slighty worse outcome in predicting the likelihood of pancreatitis diagnosis that lipase levels.

My thoughts are that firsty in the diagnosis of pancreatitis CT is not routinely used, but is used in monitoring severity and complications. Secondly, why use another diagnostic tool when a cheap blood test is accurety and available.

Although the manuscript is well-written and interesting, i can not not point the oveall weak merit.

Author Response

Radiomics Signature of Acute Pancreatitis in CT (Manuscript Nr.: diagnostics-2880562)

We would like to thank the editor-in-chief, the editors and the reviewers for their effort and highly appreciate the constructive character of their criticism and comments. The manuscript was revised as suggested and resubmitted with the following point by point response to the comments of the reviewers. We believe that the quality of our manuscript substantially improved during the review process.

Response to Reviewer 1:

Congratulations!

The presented study is evaluating an automatic machine learning model for diagnosis of acute pancreatitis using radiomics analysis. They enrolled Patients with abdominal pain and contrast-enhanced CT of the abdomen in an emergency setting. The logistic regression model showed slighty worse outcome in predicting the likelihood of pancreatitis diagnosis that lipase levels.

My thoughts are that firsty in the diagnosis of pancreatitis CT is not routinely used, but is used in monitoring severity and complications.

Secondly, why use another diagnostic tool when a cheap blood test is accurety and available. Although the manuscript is well-written and interesting, i can not not point the overall weak merit.

We thank the reviewer for consideration of our manuscript and the valuable comments. We totally agree with the reviewer that CT is not routinely used for primary diagnosis of pancreatitis, but more for detection of complications and for diagnosis of unclear cases.

We therefore emphasized this issue in the limitations section and added the following paragraph:

“Third, it is important to mention that this AI-model might be of limited value in clinical routine. The majority of cases with AP have typical clinical symptoms and elevated lipase levels and therefore, no CT is recommended for primary diagnosis. Contrast-enhanced CT is performed either in unclear cases or for suspected complications of AP. Therefore, this model might be limited for a small group of unclear cases, either in patients with typical clinical symptoms and normal lipase levels or vice versa. Further, there is also a rare number of cases with only subtle changes in CT imaging that might benefit from this quantitative model.”

We also pointed out in the other sections that the clinical use might be limited for unclear cases. However, we also believe that in the near future automatic image analysis will also be implemented in clinical routine; therefore, this work shows that automatic segmentation and texture analysis is able to diagnose acute pancreatitis comparable to the gold standard. We added:

“However, with the increasing use of AI-based detection of pathohological changes, this study provides evidence that texture-based analysis provides comparable diagnostic confidence to blood tests and may therefore be implemented in automated detection in the future.”

Reviewer 2 Report

Comments and Suggestions for Authors

This research focuses on employing machine learning for the detection of acute pancreatitis using computed tomography. Several concerns are identified in the current work:

1.       Title: The title should include the term "machine learning" to accurately convey the study's objective.

2.       Introduction: The significance of CT imaging in acute pancreatitis detection needs clarification, emphasizing its importance in the overall process. Specifically, the authors should underscore the significance of CT segmentation (texture analysis) in AP detection.

3.       Introduction: A brief introduction to radiomics in CT is recommended.

4.       Line 76: Replace "no pancreatitis" with "non-pancreatitis."

5.       Line 103: Provide information on the current (mA) in the technical parameters.

6.       Section 2.4: Elaborate on the machine learning method chosen and explain the rationale behind selecting this method over other algorithms.

7.       Sections 2.1 and 3.1: Consider combining these sections as they are related.

8.       Table 1: The presentation is unacceptable, with overlapping table content and line numbers. It appears the authors overlooked the final version before submission.

9.       Figure 2 and Figure 3: Unacceptable due to the small font size in the text on the right. This issue needs to be addressed.

1.   Table 2: Include short descriptions for all features presented in the table.

1.   Figure 5: The figure caption should explicitly mention all subfigures of A, B, and C for clarity.

Comments on the Quality of English Language

NA

Author Response

Response to Reviewer 2:

This research focuses on employing machine learning for the detection of acute pancreatitis using computed tomography. Several concerns are identified in the current work:

We thank the reviewer for consideration of our manuscript and the valuable comments.

  1. Title: The title should include the term "machine learning" to accurately convey the study's objective.

We thank the reviewer for this suggestion and changed the title accordingly: Radiomics-based Machine Learning Model for Diagnosis of Acute Pancreatitis in CT.

  1. Introduction: The significance of CT imaging in acute pancreatitis detection needs clarification, emphasizing its importance in the overall process. Specifically, the authors should underscore the significance of CT segmentation (texture analysis) in AP detection.

We totally agree with the reviewer; as already suggested by Reviewer 1 we added a further paragraph in the limitations section to point this out:

“Third, it is important to mention that this AI-model might be of limited value in clinical routine. The majority of cases with AP have typical clinical symptoms and elevated lipase levels and therefore, no CT is recommended for primary diagnosis. Contrast-enhanced CT is performed either in unclear cases or for suspected complications of AP. Therefore, this model might be limited for a small group of unclear cases, either in patients with typical clinical symptoms and normal lipase levels or vice versa. Further, there is also a rare number of cases with only subtle changes in CT imaging that might benefit from this quantitative model.”

  1. Introduction: A brief introduction to radiomics in CT is recommended.

Thank you for this comment; we added a brief introduction to radiomics.

  1. Line 76: Replace "no pancreatitis" with "non-pancreatitis."

Thank you, we changed this accordingly.

  1. Line 103: Provide information on the current (mA) in the technical parameters.

Thank you, we added further technical parameters as suggested.

  1. Section 2.4: Elaborate on the machine learning method chosen and explain the rationale behind selecting this method over other algorithms.

We added further clarification why we chose this method.

  1. Sections 2.1 and 3.1: Consider combining these sections as they are related.

We agree with the reviewer that there is some overlap between these two sections; however, we aimed to describe inclusion and exclusion criteria in the Methods section and the patient selection process in the Results section.

  1. Table 1: The presentation is unacceptable, with overlapping table content and line numbers. It appears the authors overlooked the final version before submission.

We apologize for this mistake and corrected it accordingly.

  1. Figure 2 and Figure 3: Unacceptable due to the small font size in the text on the right. This issue needs to be addressed.

Thank you, this was changed accordingly.

  1. Table 2: Include short descriptions for all features presented in the table.

We thank the reviewer for this suggestion; we added explanations for feature groups (e.g. shape, firstorder) in the Table caption to further describe these features.

  1. Figure 5: The figure caption should explicitly mention all subfigures of A, B, and C for clarity.

Thank you, we added further clarification.

Round 2

Reviewer 2 Report

Comments and Suggestions for Authors

I am satisfied with the corrections and modifications made as per my comments.

Comments on the Quality of English Language

No Comment.